# Fabrication of Carboxylmethyl Chitosan Nanocarrier via Self-Assembly for Efficient Delivery of Phenylethyl Resorcinol in B16 Cells

**DOI:** 10.3390/polym12020408

**Published:** 2020-02-11

**Authors:** Pei Zhang, Huixia Guo, Chenguang Liu

**Affiliations:** 1College of Marine Life Sciences, Ocean University of China, Qingdao 266003, China; zhangpei8877@126.com; 2Department of Life Science, Luoyang Normal University, Luoyang 471022, China; Ghuixia8@163.com

**Keywords:** carboxymethyl chitosan, self-assembly, nanocarrier, phenylethyl resorcinol, cellular uptake, hyperpigmentation

## Abstract

Micro-molecular drugs have special advantages to cope with challenging diseases, however their structure, physical and chemical properties, stability, and pharmacodynamics have more requirements for the way they are delivered into the body. Carrier-based drug delivery systems can circumvent many limited factors of drug delivery and increase their bioavailability. In this context, stable drug nanocarriers of alkaline amino acids (arginine, Arg) modified conjugated linoleic acid-carboxymethyl chitosan (CLA-CMCS) conjugate were developed, which could generate supramolecular micelles to effectively encapsulate the tyrosinase inhibitor phenylethyl resorcinol (PR). The resulting CCA-NPs were spherical nanoparticles with a mean size around 175 nm. The 3-(4,5-dimethyl-2-thiazolyl)-2,5-diphenyl-2-H-tetrazolium bromide (MTT) assay and cellular uptake investigation demonstrated that the CCA-NPs were non-cytotoxic and had excellent cell transport ability. In addition, these CCA-NPs were able to effectively deliver PR and inhibited melanin formation to reduce pigmentation by enhancing cellular uptake. In conclusion, our research indicated that nanocarriers based on self-assembly amphiphilic polymers constituted a promising and effective drug delivery system in hyperpigmentation targeting.

## 1. Introduction

Skin is one of the most common cancer sites in the human body and the degree of pigmentation is the most commonly used predictor of skin cancer risk in the general population [1]. Skin pigmentation is caused by the massive synthesis of melanin in pigment-producing cells, the melanocytes, followed by transport of melanin particles to adjacent keratinocytes [2]. Especially, hyperpigmentation is an important disease in the skin care industry, mainly due to prolonged exposure to ultraviolet light and excessive use of certain chemical drugs [3,4]. The common and effective treatments for all types of hyperpigmentary disorders are topical tyrosinase inhibitors and antimelanogenesis compounds, such as hydroquinone (HQ), kojic acid (KA), azelaic acid (AA), and phenylethyl resorcinol (PR) [5,6,7,8]. Specially, PR is an effective skin lightening agent compared with other commonly used skin lightening drugs and proved to be a highly safe ingredient [9]. However, the medical application of PR is greatly limited due to its low water solubility and light instability [10]. Nowadays, the use of nanomaterials in micro-molecule drug delivery systems is expected to break through these limitations and bring new hopes for treatment [11]. At the same time, some of these nano drug-loading systems have entered clinical trials [12]. Then, the ensuing problem is the growing demand to establish efficient and functional nanocarriers to deliver PR in cells [13].

Among the various nanomaterials for preparing drug delivery systems, amphiphilic polysaccharide derivatives have been widely used in biomedical applications such as efficient encapsulation and targeted delivery of drugs, because of their good cytocompatibility and biodegradability [14,15,16,17]. It has been found that amphiphilic block or graft copolymers can form various types of self-assembled, stable, nano-sized micellar aggregates in aqueous solution [18,19]. Due to their biosafety and biodegradable character, self-assembled nanoparticles such as lipid, polylactic-co-glycolic acid (PLGA), polyamidoamine (PAMAM) dendrimers and polysaccharide based nanoparticles consisting of an internal hydrophobic core and an externally surrounded hydrophilic group, have been widely reported to increase the delivery efficiency and bioavailability of drugs [20,21,22,23]. Among these nanoparticles prepared from polymers, there has been rising interest in nanoscale self-aggregates of natural polysaccharides such as pullulan [24], curdlan [25], dextran [26], alginic acid [27], and chitosan [28]. Carboxymethyl chitosan (CMCS) is one of the water-soluble derivatives of chitosan, which has attracted the most attention and exploration in the field of biomedicine [29]. Many researchers have focused on CMCS as a source of potential bioactive material and the nanocarriers-based on CMCS have been rapidly developed in recent years [30,31,32]. The entrapment of micro-molecular drugs in CMCS has been reported, including gefitinib [33], resveratrol (RES) [34], levofloxacin [35], doxorubicin [36], nepafenac [37], glycyrrhizic acid (GA) [38], etc. Specifically, spherical RES:CMCS-NPs were prepared under optimal conditions, in which average particle size, potential, drug loading and encapsulation efficiency were (155.3 ± 15.2) nm, (−10.28 ± 6.4) mV, (5.1 ± 0.8)% and (44.5 ± 2.2)%, respectively. In addition, the crosslinked GA-loaded CMCS-NPs dissociated to release drug in the presence of glutathione at a concentration comparable to the intracellular environment, featuring the potential ability of this system for intracellular delivery.

In this work, we present a nanocarrier formed with the amphiphile hydrophobic-modified CMCS conjugate, which contained a hydrophilic polysaccharide skeleton (CMCS), a hydrophilic functional amino acid chain (Arg) bridged via a newly-formed amide bond, and a hydrophobic fatty acid chain (CLA) bridged via a newly-formed ester bond. CLA had various functions such as inhibiting the formation of cancer and tumors, losing weight, anti-atherosclerosis, improving immune function, reducing cholesterol, and promoting growth. Arginine, which has been widely used as a cell penetrating peptide mediating transepithelial transport, would be grafted to CLA-CC block copolymer by amide bond. The successful preparation of CCA nanoparticles was demonstrated and their physicochemical properties were characterized, such as mean size, PDI, and zeta potential. Cytotoxicity and cellular uptake profile of these nanoparticles were also investigated to determine the optimum conditions. In addition, the resulting PR:CCA-NPs can effectively enhance the encapsulation capability of PR and increase its cellular uptake in vitro.

## 2. Materials and Methods

### 2.1. Materials

Carboxymethyl chitosan (CMCS, substitution degree of carboxymethyl >80%, viscosity = 600 mpa s) was obtained from Heppe Biotechnology Co.Ltd. (Qingdao, China), Phenylethyl resorcinol (PR) was acquired from Tokyo Chemical Industry Co.Ltd. (Tokyo, Japan). Conjugated linoleic acid (CLA), 1-ethyl-3-(3-dimethylaminopropyl) carbodiimide (EDC), N-hydroxysuccinimide (NHS), DMEM medium were from Solarbio Co.Ltd.. MTT, L-Arg, DiD labeling solution and Dialysis tube (Molecular weight cut off for targeted drug delivery: *M_W_*CO = 8000 ~ 14,000 D) were obtained from Sigma Chemical Co. Ltd. (St. Louis, MO, USA). Cell culture flasks (25 cm) and plates were from Costar Co.Ltd. (New York, NY, USA) The mouse fibroblast cell line L929 and mouse B16 melanoma cells were provided by the Shanghai cell bank.

### 2.2. Synthesis of CLA-CMCS-Arg Conjugates

Herein, the CMCS conjugates are prepared by the methods previously described with modifications [18]. Firstly, the CMCS was completely dissolved in 100 mL of distilled water (2%, *w*/*v*) and the CLA was dissolved in 85 mL of methanol, and then these two solutions were mixed. Subsequently, 15 mL of EDC methanol solution was added dropwise to the mixture while stirring at room temperature. Secondly, the activated Arg solution (the EDC/NHS coupling reaction solution) was added to the CLA-CMCS mixture in a certain ratio. The resultant solution was stirred for 24 h at room temperature and then dialysised for 2 days. Finally, the mixture was freeze-dried to obtain the CCA products named CCA1-5. The prepared CCA conjugates was characterized by fourier transformed infrared spectroscopy (FTIR) spectrum and ^1^H nuclear magnetic resonance (NMR) spectroscopy.

### 2.3. Preparation and Characterization of CCA-NPs and PR-Loaded CCA-NPs

CCA-NPs were prepared using ultrasonic induced self-assembly technology. In details, the CCA conjugate was sufficiently dissolved in distilled water at 25 °C for 24 h. The solution was then sonicated 3 times for 5 min using a probe-type sonicator (Sonics Ultrasonic Processor, VC750, Hartford, CT, USA) at a frequency of 200 W, with the pulse being turned off at 2.0 s for 2.0 s to prevent temperature rise [39].

Morphological examination of CCA-NP was performed by TEM (H-600A, Hitachi, Japan). The sample was diluted with water and placed on a copper mesh, which were air dried and negatively stained with 1% (*w/v*) phosphotungstic acid before observation. CCA-NPs were dispersed in 0.2 M PBS (pH 7.4) for determining their particle size, polydispersity (PDI) and zeta potential by the Malvern Zetasizer 3000HSA (Malvern, UK) at room temperature.

The CCA-NPs were used to encapsulate PR via ultrasonic dispersion method [11]. After centrifugation of the aqueous dispersion, the percentage of PR (encapsulation efficiency) was determined by spectrophotometry at 280 nm using a Biorad 680 microplate reader. The amount of free drug was detected in the supernatant, and the amount of drug incorporated was determined by subtracting the free drug from the initial drug. The PR encapsulation efficiency (EE%) was calculated by the following equation:EE (%) = *W*_t_/*W*_o_ × 100%
where “*W*_t_” represents the amount of PR that loaded into nanoparticles, “*W*_o_” represents the initial amount of PR fed.

In order to label CCA-NP and PR:CCA-NP with fluorescein, these NPs were suspended in PBS buffer, followed by addition of a solution of FITC in absolute ethanol and the mixture was magnetically stirred at room temperature for 24 h [40]. The FITC-NPs suspension was then dialyzed against ethanol for three days and a new ethanol solution was replaced daily to remove unlabeled FITC. The progress of this experiment and the preservation of the labeled product should be in the dark.

### 2.4. Cell Experiment

The L929 and B16 cells were cultured in Dulbecco’s modified Eagles’s medium supplemented with 10% heated-inactivated fetal bovine serum, 100 U/mL penicillin, 2 mM glutamine and 100 μg/mL streptomycin. In order to ensure the accuracy and scientificity of cell results, all cell experiments such as culture, seeding, and cytotoxicity were performed under the same conditions. L929 and B16 cells were seeded at a concentration of 1 × 10^4^ cells/mL into a 96-well plate for further attachment and growth [41]. Following the cultivation, the DMEM medium were changed with new complete medium containing CCA-NPs with different DS at the concentrations of 250–2000 μg/mL. The treated cells were incubated for 1, 2, 4, 12, and 24 h at 37 °C and 5% CO_2_, maximizing the uptake of the particles by cells. The cytotoxicity of NP was measured using the MTT method, and the absorbance of each well at a wavelength of 492 nm was measured using a microplate reader. Cell viability is expressed as a percentage compared to the control group. The cellular uptake of FITC-CCA-NPs was visualized by fluorescence microscopy (80i, Nikon, Japan) and quantified by fluorescence intensity given by a fluorescence microplate reader (FLx800B, Bio-Tek, Burlington, VT, USA). Specially, a part of the cultures was stained with DiD to visualize membrane morphology. Cultures were stained for 10 min with DiD, washed with DMSO and PBS and were viewed under a microscope.

### 2.5. Detecting the Content of Melanin

The method for determining the content of melanin was slightly modified as reported previously [42]. B16 cells were seeded at a concentration of 1 × 10^4^ cells/mL into 96-well plates and incubated overnight to allow cell adhesion. Cells were treated with PR-loaded CCA-NP at various concentrations (20, 50, and 100 μg/mL) for 48 h. After washing the cells with PBS, we inverted the cell plates and tapped on a paper towel to pour out the entire wash solution. Then, cells were lysed with 1 mL of 10% DMSO in NaOH solution for 1 h at 80 °C. Subsequently, each crude cell extract (200 mL) was transferred to a new 96-well plate and the relative density of melanin was calculated by measuring the optical density at a wavelength of 400 nm using a microplate reader. The final content of melanin was calculated from a standard curve drawn from a melanin standard purchased from Sigma.

### 2.6. Statistical Analysis

In this study, all data were analyzed statistically by the SPSS 24.0 software (IBM, Armonk, NY, USA) package and reported as mean ± standard deviation. Chemical structures were drawn using ChemBioDraw Ultra 14.0 software (Cambridge, MA, USA).

## 3. Results and Discussion

### 3.1. Characterization of CCA Sample

The amphiphilic CCA conjugates prepared in this work (Figure 1) contain a hydrophilic polysaccharide skeleton, a hydrophilic functional amino acid chain bridged via a newly-formed amide bond, and a hydrophobic fatty acid chain bridged via a newly-formed ester bond [43]. Different CCA conjugates were prepared by varying the feed ratio of CLA and L-Arg to CMCS (Table 1). In this part, EDC is a cross-linking agent called “zero length” that provides an amide bond without leaving a spacer molecule. EDC could react with the carboxyl group of the CMCS to produce an active ester intermediate, which could chemically couple with Arg-NH_2_. However, EDC is not very stable in water because the oxygen atoms act as a nucleophile and inactivate the cross-linker agent. Using NHS is a preferable way to improve stability of EDC. It is known the most efficient coupling reactions occur in acidic conditions [44]. A schematic representation of the CCA interactions, mainly the covalent bond interaction and hydrogen bond interaction, is shown in Figure 1.

The characteristic absorption peaks of CMCS, CLA-CMCS and CLA-CMCS-Arg were analysed using FTIR spectroscopy, which are shown in Figure 2A, indicative of their interaction. The peaks at 1591 cm^−1^, 1414 cm^−1^, 1070 cm^−1^ and 2928 cm^−1^ are all characteristic peaks of CMCS, assigned to the stretching vibrations of C=O, CH_2_COO–, –C–O– group and the tensile vibration of C–H separately [45,46]. Most importantly, the augmented peak at 1736 cm^−1^ corresponding to the stretching vibration of C=O group indicated the existence of the C=O group of a newly-formed ester bond in CC polymer [47] and escalated peaks at 1640 cm^−1^ stands for CO stretching vibration and indicated the formation of an amide bond linkage between Arg and CMCS [48]. Similarly, ^1^H NMR spectroscopy (Figure 2B) further demonstrated the formation of new ester and amide bonds in CCA conjugate. The chitosan backbone peaks present at 3.0, 3.5–3.8, and 4.2 ppm are assigned to the carbon d, carbon a–c,f, and carbon e of chitosan, respectively [49]. The new peak indicates that the CLA and Arg groups are directly linked to the CMCS backbone. The new peaks at 0.7–1.0, 1.0–1.3,2.1, 2.5, 4.6, 5.0 and 5.5 ppm were attributed to the pendant groups of CLA, and the new peaks at 1.7, 1.9, 3.2–3.3, 4.6 ppm were attributed to the groups of Arg [18]. In conclusion, the FT-IR and ^1^H NMR results demonstrate the successful synthesis of CCA.

### 3.2. Formation and Characteristics of Self-Aggregated Nanoparticles

CCA nanoparticles can be easily formed by ultrasound-assisted self-assembly, which can be characterized by dynamic light scattering detection (DLS, Table 1) and transmission electron microscopy (TEM, Figure 3). The so-formed CCA1 assemblies were spherical in shape (203.4 ± 3.42 nm in diameter) and dispersed polydispersity index (PDI:0.252) which demonstrated a narrow particle size distribution. In addition, the zeta potential is −39.7 ± 0.26 mV, which may be due to the entanglement of some CMCS molecules on the surface of the particles to produce a negative charge. Absolute value of zeta potential was greater than 30mV indicated that the seprepared colloidal system was stable [50]. We speculated that CCA-NPs exhibited a spherical core/shell architecture, with functional groups on the surface and inner hydrophobic cores, respectively.

To more intuitively understand the shape and size of the CCA-NP, the morphology of the nanoparticles was evaluated using TEM. The transmission electron micrograph of CCA1 and CCA5 nanoparticles was shown in Figure 3. The NPs were spherical in shape with smooth surface, and have narrow size distribution. Specifically, CCA5-NPs exhibited non-circularity and irregular shape and the particle size also increased. This result was attributed to the fact that L-Arg hydrophilic chain on the polysaccharide backbone is entangled outside the core in a relatively loose state and stretched outward of the hydrophilic group causes the entire particle to loosen and swell. Also of note is that the particle size observed by the TEM image (Figure 3) was in the range of 160–200 nm, which seems to be smaller than the result obtained by DLS (Table 1). The difference in size could be reasonably ascribed to the hydrodynamic radius in DLS [49]. In addition, Morphological examination of CCA5-NP and PR:CCA5-NPs were performed by TEM. The mean sizes of blank CCA-NPs and PR:CCA-NPs were found to be 206.5 ± 3.28 nm and 237.4 ± 1.94 nm, repectively. The increase of particle size of PR:CCA-NPs as compared to that of blank CCA-NPs could be due to the loading of PR in the hydrophobic core of the nanoparticles [11].

Table 1 and Figure 3 show the particle size distribution and particle shape of CCA conjugate in PBS solution. As we can see, the sizes of a majority of nanoparticles are 190–230 nm which are significantly affected by increasing CLA and L-Arg to CCA conjugates. Similarly, cholesteryl-bearing pullulan (CHP) with the higher DS of cholesterol formed more densely packed particles due to the increase of the cholesterol number in one hydrogel nanoparticle [51]. We further investigated encapsulation efficiency of the CCA-NPs to find that addition of CMCS significantly increased the EE of PR. The highest EE (81.93 ± 5.01) was obtained at ratio 4:4:1 of CLA: CMCS: Arg. As shown above, the distribution of PR and hydrophobic chains is completely identical, which confirms that PR is mainly encapsulated in the hydrophobic core of the nanoparticles. Therefore, these results confirm that the increase in its inner core can achieve the high PR payload [18].

### 3.3. Cytotoxicity and Cellular Uptake of CCA-NPs

Cytotoxicity is an unavoidable problem for most polymeric drug carriers such as poly (L-lysine). It was reported that the cytotoxicity of polymers may have serious adverse effects on cell membranes and extracellular matrix proteins, thereby reducing drug penetration [52]. The toxicity of CCA5-NP to L929 cells was assessed using the MTT method. The cell viability of all groups was above 80% when the concentrations increased from 250 to 1500 μg/mL (Figure 4A), which demonstrated that the CCA-NPs were nontoxic in a suitable concentration range. In addition, there was no significant difference in cell viability between the five groups of CCA-NPs, indicating that the modification did not affect the excellent biosafety of chitosan and the excellent biocompatibility of the CCA-NPs was favorable for PR delivery.

Nowadays, the successful application of NPs in drug delivery requires efficient cellular uptake, which depends on special carriers that can recognize and transport L-Arg on cell membrane [53]. As shown in Figure 4, cellular uptake efficiency increased with extended incubation time changed from 1 h to 4 h (Figure 4B) and the larger material ratio of L-Arg (CCA5, Figure 4C). Obviously, CCA-NPs were good nanocarriers for penetrating cell membranes and drug delivery, depending on the time of cell penetration and arginine content on the surface of NPs (Figure 4D). We inferred that cellular uptake of CCA-NPs was an energy-dependent and saturable process with the possible cellular transport route of clathrin. In the future, clathrin, caveolae, and macropinocytosis inhibitors could be used in further studies to elucidate the specific transport mechanism of CCA-NPs. Also, the data about the effects of internal factors and external environment on the cellular uptake efficiency provided more information on how to achieve the best delivery efficacy [54].

### 3.4. B16 Cells Death Induced by PR:CCA-NPs

Impelled by the favorable higher PR EE and cellular uptake efficiency of CCA-NPs (Table 1 and Figure 4D), we continued to evaluate their inhibition rate in B16 cells. Cytotoxicity in B16 was monitored using the MTT assay, and microscopic observations were carried out as indicators for cell viability. As shown in Table 2, the proliferation inhibition of B16 cells was enhanced with the increase of Free PR concentration and the prolongation of treatment time. After treatment with 200 μg/mL free PR for 48 h, the inhibition rate of B16 cell proliferation reached 79.42 ± 1.27%. It suggested that the inhibit proliferation of free PR was time and concentration dependent in a certain range, and we selected the appropriate four concentrations of free PR (10–100 μg/mL) for the encapsulation and cytotoxicity of PR:CCA-NPs.

The effect of four different concentrations of PR: CCA1-NPs and three different drug-loaded nanoparticles on cytotoxicity is shown in Figure 5. The cell viability is significantly lower than 75% when the drug concentration is 50 μg/mL which indicates that it has an inhibitory effect on the proliferation of B16 cells (Figure 5A). Specifically, the cell proliferation rate of PR:CCA1-NPs (78 ± 3.06%) is much higher than these of free PR (62 ± 5.41%) at the same concentrations (50 μg/mL) (Figure 5B). These data directly indicated that PR:CCA-NPs were more efficient than free PR in reducing drug sensitivity. Analyzed by observation after treatment, the B16 cells of control group were full and radial with complete cell membrane, good refractive index and rapid proliferation (Figure 5C–E). After 24 h of treatment with PR: CCA-NPs, a large number of cells degraded, the cell membrane ruptured, and the cell adhesion ability was poor, showing a massive death (Figure 5F–H). Evaluating the nature of cell death induced by NPs, the results suggested that the PR:CCA-NPs induced a necrotic cell reaction [52]. In the future, apoptosis and autophagy which did not detect any indication in this part could be researched in further studies to elucidate the specific death mechanism.

To assess the efficacy of CCA-NPs for in vitro applications, we performed the uptake assay of FITC-labeled PR:CCA5-NPs by B16 cells. A more detailed performance analysis of FITC-labeled nanoparticles had been reported in our previous work [40]. After the PR:CCA5-NPs sample was added to B16 cells and continued to incubate for four hours, the cell uptake images were taken with a fluorescence microscope. B16 cells had good uptake profile of PR-loaded nanoparticles (Figure 6A–C), however, the uptake reached a steady state after treatment for 12h. Notably, the modified drug-loaded nanoparticles remained intact and stable after storage for one month at 4 °C (Figure 6E), further confirming their robust nature and practicality of CCA polymer.

The direct cause of skin pigmentation is an increase in melanin production, and it is based on the degree of inhibition of cellular melanin synthesis to evaluate whether PR is effective [55]. It can be seen from the inhibition results that the three nanoparticles can inhibit the synthesis of melanin after PR loading, and the effect is most obvious in the concentration of 50 μg/mL (Figure 6D). In all cases, the inhibition effect of PR:CCA-NPs on melanin production was much higher than that produced by the free PR. These conclusions are related to the cell proliferation described previously, reducing the production of melanin by PR:CCA-NPs while causing cell death. In addition, it could be found that CCA5 nanoparticles with the highest Arg content promoted cell uptake efficiency and delivered more PR drugs, finally resulting in a stronger inhibition of melanin production (Figure 7).

## 4. Conclusions

In this study, we developed a simple synthesis of well-distributed spherical CCA-NPs synthesized from the amphiphilic CMCS polymer, which provides efficient encapsulation and delivery of the tyrosinase inhibitor PR and inhibits melanin formation while obviating systemic toxicity. The CCA-NP prepared in this work had a uniform solution without any aggregation in the one-month period at room temperature. The hydrophobic cores enhanced the loading of the inhibitor PR in CCA particles which showed low cytotoxicity. The PR: CCA-NPs had multiple therapeutic advantages for inhibiting melanogenesis, including: (i) efficient encapsulation; it was mainly attributed to its unique amphiphilic structure which caused the nanoparticles to have more low-density hydrophobic cores capable of containing large amounts of drugs after self-assembly; (ii) small particle size; this small size of nanoparticles could quickly deliver more drugs to tumor cells and prolong the beneficial accumulation time in cells through sustained release; (iii) efficient cellular uptake of NPs mainly via its transmembrane groups and boosted receptor-mediated endocytosis; (v) effectively improved the efficacy of drugs.

With so many advantages as described above, CCA-NPs could effectively enhance the delivery of PR and inhibitory effect of melanin formation in vitro. In addition, CCA-NPs could reduce the cytotoxicity of PR by adding more functions and targeting groups to the polysaccharide backbone and control the release effect of PR. Based on these interesting and scientific results, this amphiphilic CMCS polymer-based drug delivery system is expected to be a feasible solution for hyperpigmentation targeting. Further studies are warranted to elucidate the underlying molecular mechanisms of action involved in melanogenesis inhibition and the design of animal experiments to verify the suitability of drug-loading systems for skin lightening in vivo.

## Figures and Tables

**Figure 1 polymers-12-00408-f001:**
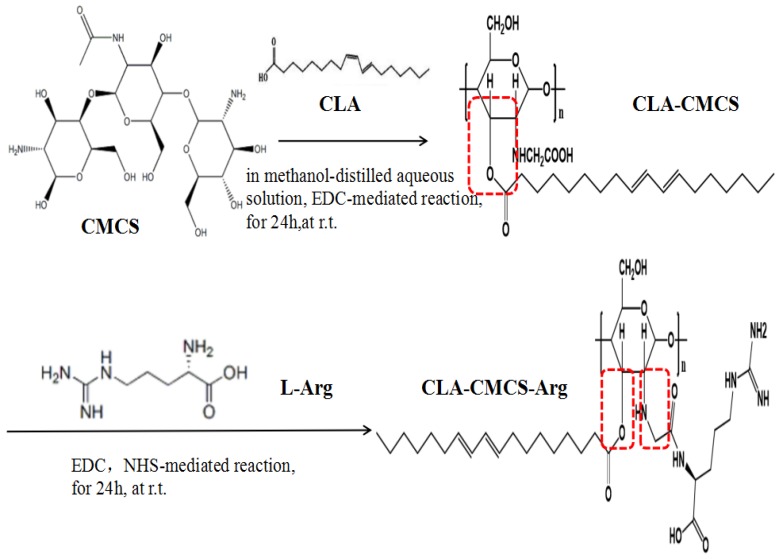
The schema of CLA-CMCS-Arg (CCA) synthesis.

**Figure 2 polymers-12-00408-f002:**
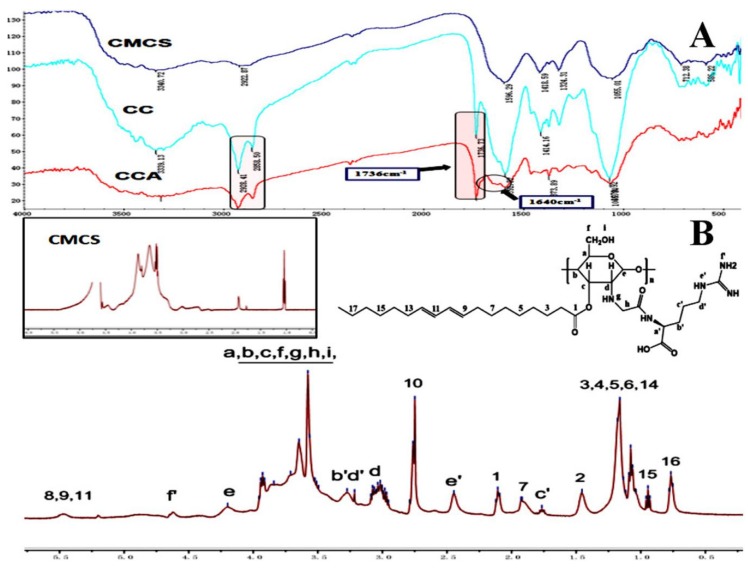
FTIR (**A**) and ^1^H NMR (**B**) spectra of CLA-CMCS-Arg.

**Figure 3 polymers-12-00408-f003:**
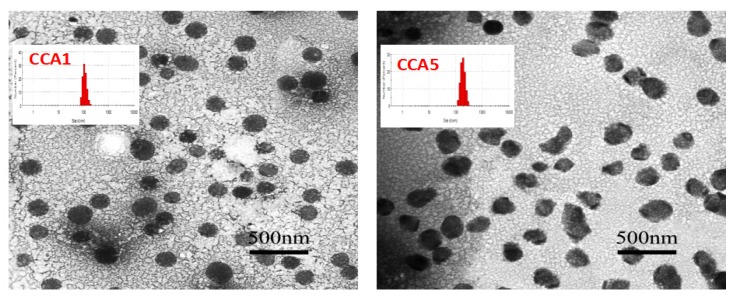
TEM of CCA1 and CCA5 nanoparticles.

**Figure 4 polymers-12-00408-f004:**
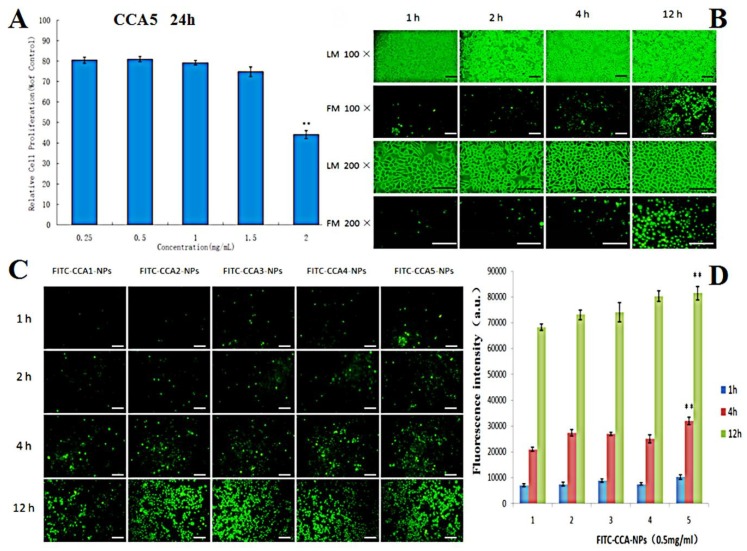
(**A**) Effects of CCA5-NPs on the proliferation ability of L929 cells (24 h), ** *p* < 0.01:2 mg/mL vs other concentrations; (**B**) Light microscopy (LM) and fluorescence microscope (FM) images of L929 cells treated with CCA5-NPsand FITC-CCA5-NPs: 1 h, 2 h,4 h and 12 h, the scale bar represents 25 µm; (**C**) FM images of L929 cells treated with FITC labeled CCA(1–5)-NPs, the scale bar represents 25 µm; (**D**) Cellar uptake efficiency of CCA(1–5) nanoparticles by L929 cells at various time(1,2,4 and 12 h), ** *p* < 0.01: 12 h vs 1 h and 4 h (CCA5-NPs).

**Figure 5 polymers-12-00408-f005:**
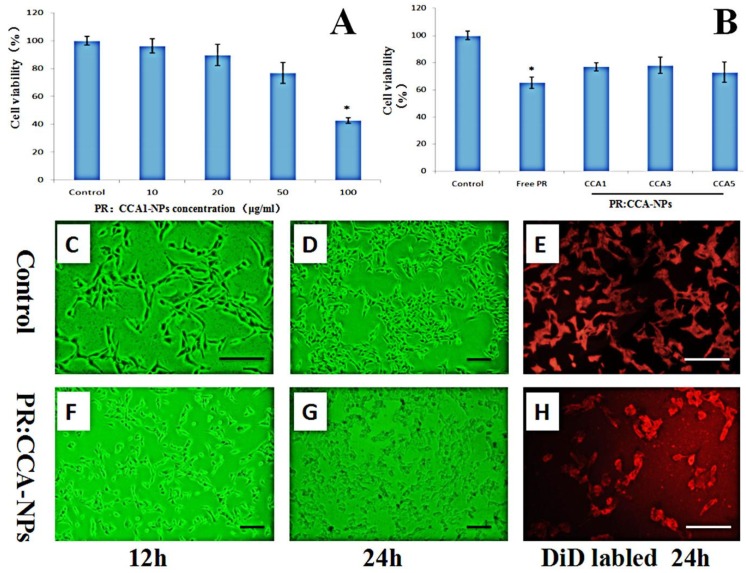
Effects of different concentrations of nanoparticles (**A**) and different PR:CCA-NPs (**B**) on the proliferation ability of B16 cells (24 h); Observing the B16 cells of mice at different time under inverted microscope (**C**, **D**, **F**, **G**); Cells were stained with DiD to visualize membrane morphology (**E**, **H**). Note: the scale bar represents 25 µm; * *p* < 0.05 vs control.

**Figure 6 polymers-12-00408-f006:**
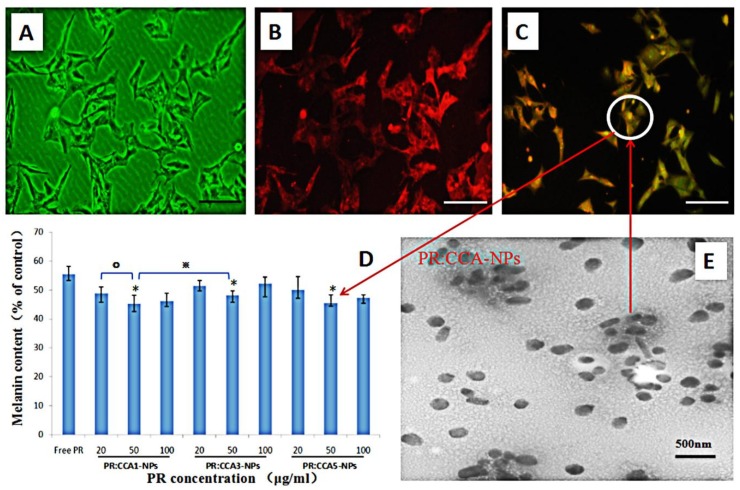
(**A**–**C**) The FITC labled PR:CCA-NPs were added to B16 cells and co-incubated for 4h to detect the effects of PR: CCA-NPs on pinocytosis by cells, the scale bar represents 25 μm. (**A**) Control group, (**B**) DiD labeled cell membrane, (**C**) FITC labeled PR:CCA-NPs; (**D**) Effects of different NPs with different concentration on cellular melanin content in B16 cells; E: TEM of PR: CCA5-NPs. Note: * *p* < 0.05: CCA-NPs (50 μg/mL) vs Free PR; ^※^
*p* < 0.05: CCA1 (50 μg/mL) vs CCA3 (50 μg/mL); ° *P* < 0.05:CCA1 (50 μg/mL) vs CCA1 (20 μg/mL).

**Figure 7 polymers-12-00408-f007:**
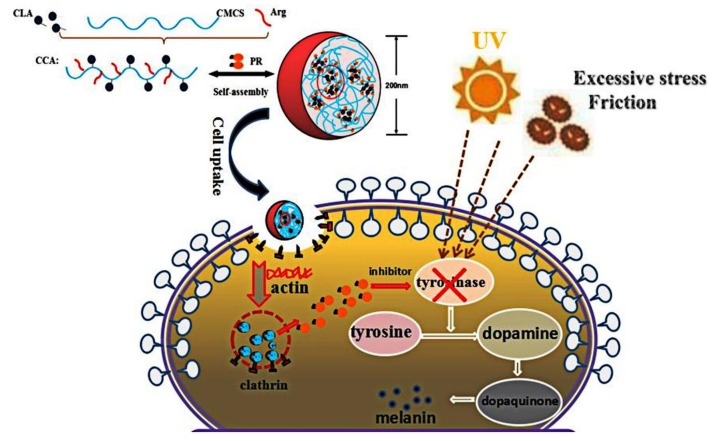
Diagram of self-assembly of CCA5-NPs in distilled water and efficient delivery efficiency of PR in B16 cells.

**Table 1 polymers-12-00408-t001:** General properties of CCA (1–5) conjugates in distilled water.

Components Weight (g)				Mean Size (nm)	PDI	Zeta Potential (mV)	EE (%)
Samples	CLA	CMCS	Arg				
1	0.25	1.00	0.25	203.4 ± 3.42	0.252	−39.7 ± 0.26	69.21 ± 1.96
2	0.50	1.00	0.25	192.4 ± 6.18	0.192	−42.8 ± 1.37	71.48 ± 2.69
3	1.00	1.00	0.25	193.1 ± 2.69	0.218	−47.4 ± 2.34	81.93 ± 5.01 *
4	0.25	1.00	0.50	210.2 ± 3.08	0.286	−36.2 ± 1.52	76.71 ± 2.69 *
5	0.25	1.00	1.00	229.4 ± 1.83	0.230	−33.9 ± 0.43	75.15 ± 3.17 *

Note: * *p* < 0.05 vs Sample 1.

**Table 2 polymers-12-00408-t002:** Inhibition rate of Free PR on B16 cell proliferation (n = 6, $\bar{x}$
± s).

Concentration (μg/mL)	Inhibition Rate/%	
	24 h	48 h
0	0	0
5	−0.42 ± 0.52	−1.07 ± 0.26
10	8.41 ± 1.08	12.16 ± 1.37
20	24.19 ± 2.69	32.54 ± 2.34
50	42.93 ± 3.08	52.24 ± 1.52
100	60.94 ± 1.83	69.51 ± 0.43
200	65.43 ± 0.98	72.92 ± 1.27
500	73.14 ± 2.18	79.42 ± 0.49

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
