# Peer review of "Fabrication of Carboxylmethyl Chitosan Nanocarrier via Self-Assembly for Efficient Delivery of Phenylethyl Resorcinol in B16 Cells"

_polymers, 2020, doi:10.3390/polym12020408_

Round 1

Reviewer 1 Report

The authors thoroughly addressed the issues from all reviewers' comments and I recommend that this work would be accepted. 

Author Response

Dear reviewer,

Thank you for your work to deal with our manuscript titled “Fabrication of carboxylmethyl chitosan nanocarrier via self-assembly for efficient delivery of phenylethyl resorcinol in B16 cells” . 

Now, I provide the revised manuscript and cover letter to previous review comments of “Polymers-711661”. We thank you for the valuable comments and suggestions and we followed all of suggestions made. These comments helped us to improve our manuscript and provided important guidance for our future research.

Best regards !

Pei Zhang

Reviewer 2 Report

The authors have addressed all the comments and suggestions I made in the first review. The quality of the article has significantly improved. 

I appreciate the explanation of the NHS function, but my question was only to include the NHS in the experimental section because in the first version it was missing. There is no need to further explain it in the text.

Figure 1 is still wrong. The position of the double bonds in the linoleic acid are wrong. Please look carefully. In figure 1 CLA first appear with the double bonds in the position 2 and 4. Then, in CLA-CMCS the double bonds appear in the position 9 and 12 and they are not conjugated. So I do not know what linoleic acid the authors are using. Is it conjugated or not conjugated? In addition in figure 2 the numbering of the chain is wrong, the carbon of the carboxyl group is the number 1.

Author Response

Dear reviewer,

Thank you for your work to deal with our manuscript titled “Fabrication of carboxylmethyl chitosan nanocarrier via self-assembly for efficient delivery of phenylethyl resorcinol in B16 cells” . 

Now, I provide the revised manuscript and cover letter to previous review comments of “Polymers-711661”. We thank you for the valuable comments and suggestions and we followed all of suggestions made. Any change to our manuscript has been marked IN RED in the revised paper. These comments helped us to improve our manuscript, and provided important guidance for future research.

Best regards !

Pei Zhang

Point 1: I appreciate the explanation of the NHS function, but my question was only to include the NHS in the experimental section because in the first version it was missing. There is no need to further explain it in the text.

ResponseWe thank to the reviewer for the valuable comment and we followed all of suggestions made. According to the comment, we have added that the activated Arg solution (the EDC/NHS coupling reaction solution) was added to the CLA-CMCS mixture in a certain ratio. 

Also, we have added the reason for choosing EDC and NHS to the discussion in the Result 3.1. EDC could react with the carboxyl group of the CMCS to produce an active ester intermediate, which could chemically couple with Arg-NH2. However, EDC is not very stable in water because the oxygen atoms act as a nucleophile and inactivate the cross-linker agent. Using NHS is a preferable way to improve stability of EDC. It is known the most efficient coupling reactions occur in acidic conditions 44.(S. Kele, Temur, M. Altunbek and M. Culha, Applied Surface Science, 403, 455-463.). In summary, the entire experimental process and conclusions are more scientific and reliable .

Point 2: Figure 1 is still wrong. The position of the double bonds in the linoleic acid are wrong. Please look carefully. In figure 1 CLA first appear with the double bonds in the position 2 and 4. Then, in CLA-CMCS the double bonds appear in the position 9 and 12 and they are not conjugated. So I do not know what linoleic acid the authors are using. Is it conjugated or not conjugated?

ResponseWe are very sorry that we ignore such a serious error in the figure 1. Now, we have carefully checked and corrected the entire wrong bonds.

Figure 1 The schema of CLA-CMCS-Arg (CCA) synthesis 

Point 3: In addition in figure 2 the numbering of the chain is wrong, the carbon of the carboxyl group is the number 1.

ResponseWe are very sorry for that we provide the wrong numbers of the chain in figure 2. Now, we have corrected the figure 2 as follows:

Figure 2 FTIR (A) and 1H NMR (B) spectra of CLA-CMCS-Arg

This manuscript is a resubmission of an earlier submission. The following is a list of the peer review reports and author responses from that submission.

Round 1

Reviewer 1 Report

In the manuscript entitled “Fabrication of CMCS nanocarrier via self-assembling for efficient delivery of phenylethyl resorcinol in B16 cells” the authors claim to have developed a novel drug nanocarrier (CCA-NPs) to deliver a novel tyrosinase inhibitor phenylethyl resorcinol (PR). It's important to be careful with the words. The nanocarrier has some novelty but the tyrosinase inhibitor phenylethyl resorcinol (PR) has already been incorporated in other nanocarriers, for example in nanostructured lipids doi: 10.3390/nano7090241. Thus, the novelty of this work is the incorporation of PR in another nanocarrier. However, for the readers of this journal what will be the interest in the present methodology, as the nanocarrier synthesis is already reported with some modifications by the authors (doi:10.1016/j.colsurfb.2008.11.026)? The use of NHS to introduce the arginine is missing from the experimental procedure. It appears in Figure 1, but in the procedure, we have “the activated Arg solution”. Also, in Figure 1, the position of the double bonds in the linoleic acid are wrong. It is not clear to me why the authors choose arginine. I understood it is to increase cellular uptake, but this could be discussed in the introduction, because on this carrier and from what it is already published by the authors this has some novelty. I have some concerns about some of the claims the authors make regarding the impact of work. The overall conclusion is that they developed a promising effective drug delivery system in skin lightening. However, I think they can only conclude that they efficiently deliver PR into melanoma cells and that their nanocarrier show efficacy with the cellular tyrosinase inhibition assay. Overall, the authors need to clarify the novelty of the manuscript and improve the characterization.

Page 2, Line 68. I cannot understand “stability of lighten drug” Page 4, Line 149. I cannot understand reference 44 on that sentence. Reference 44 is a paper about dendrimers that self‐assembled in water into spherical nanostructures, but none of the molecules are the same. Page 5, Figure 1. Figure 1 need major work. Also, what is PNP? It does not appear in the entire manuscript. If the figure is copy/paste from another source the source should be indicated. Page 5, Line 169. In reference 50, there is no chitosan. So, this reference cannot support that peak assignment. Page 6, Figure 2. We can barely see the NMR scale. I think two-dimensional NMR spectra can help to support that peak attribution, because literature from other similar molecules, could not have the same chemical shift. Page 6, Line 183. As a rule of thumb, zeta potentials between -30 and +30 are said to be not stable. That is assuming that electrostatic charge is the only mechanism that stabilizes your colloidal system. If steric stabilization is also involved, this rule is no longer valid. It also depends on the mobility and in the material itself. If you want to say that the particles are stable, you must measure the size for several days and see if you have aggregates or not. Page 7, Line 194. The size observed by TEM is smaller because in DLS we have the hydrodynamic radius. Page 11, Line 288. Why green synthesis? It is the first time this appear in the manuscript.

Reviewer 2 Report

This study claimed to develop Arg modified carboxymethyl chitosan (CLA-CMCS) conjugates for delivering phenylethyl resorcinol (PR) for skin disorder. They characterized the CLA-CMCS conjugates and PR-encapsulated CLA-CMCS was employed in vitro cell experiments to verify the inhibition of skin disorder. However, I believe that some of important experimental details are missing and the novelty of this work was not clarified. Also, they used many acronyms which did not explain at the first time and this made me difficult to follow the manuscript. I also found a lot of typos which I did not correct and this should be corrected.

Carboxymethyl chitosan has been widely used as a nano-carrier for cancer therapy and arginine modified carriers is not new. What is your novelty to develop the arginine modified linoleic acid carboxymethyl chitosan for this work? What difference does it have compared to other carboxymethyl chitosan nano-carriers?  Line 40-41: You should include more references about micromolecule drug delivery system. There are tons of decent research papers in literature. Line 205: How do you confirm that PR was well incorporated into the micelles? You should include the scientific data to prove this. Other papers utilize dissipative particle dynamics to confirm density distribution. At least, you should do TEM and DLS  measurements after loading the drug in the nano-carriers for confirmation. Why did you use L929 cells for cytotoxicity and cellular uptake experiment and use B16 cells for inhibition rate experiment? Should they be the same cells for consistency? Line 249-251: Where can I find the 28% antiproliferation effect of PR:CCA1-NPs? Figure 5B shows the cell viability, but you talked about antiproliferation rate. Hard to follow. Line288: What makes your process green synthesis? Did you study the stability of your fabricated product? If you want to claim the "stability", please show the appropriate data.

Reviewer 3 Report

In ‘Fabrication of CMCS nanocarrier via self-assembling for efficient delivery of phenylethyl resorcinol in B16 cells’, Zhang et al. prepared nanoparticles of arginine modified lineolic acid-carboxymethyl chitosan encapsulating phenylethyl resorcinol. They evaluated the particle size, cytotoxicity, cellular uptake, and delivery of phenylethyl resorcinol for the nanocarriers.

Specific comments:

The manuscript should be carefully edited for grammar mistakes.

The figures in general should be reformatted so that they are more legible. In most cases, the resolution is low (grainy images) and the font size is often too small to be readable (particularly Figure 1 (CLA molecule) Figure 2, Figure 3 (insets), Figure 4A).

Title: CMCS should be written out in full; self-assembly would be more grammatically correct than self-assembling.

Line 19: How are effects on systemic toxicity shown, as no in vivo work was performed.

Line 49: Undefined abbreviation PAMAM

Section 2.3: Is it true that the nanoparticles with and without phenylethyl resorcinol were prepared in two different ways(ultrasonic self-assembly and film dispersion)? More details of the film dispersion method should be supplied. Does this also use sonication? Otherwise, how can it be assumed that the nanoparticles with and without phenylethyl resorcinol have similar properties?

Lines 168-169: It is unclear how the peaks are being assigned. There are more groups listed than the number of peaks.

Lines 169-170: What is the new peak?

Line 179: How was it determined that the CCA2 assemblies were spherical in shape. No TEM images are shown of these samples.

Lines 200-201 and 204: What is the p-value for the significant differences?

Line 201: What is CHP?

Did the size or zeta potential of the nanoparticles change after encapsulation of phenylethyl resorcinol?

What are the abbreviations LM and FM in Figure 4B?

Line 227: In Figure 4C-D, are any of the differences statistically significant? How is it concluded that addition of L-Arg improves the cellular uptake?

Lines 249-250: What data is supporting this statement? The viability of the cells treated with free phenylethyl resorcinol is lower than that of the cells treated with the nanoparticles in Figure 5B.

Line 251: What is the relevance of reference 40 for this statement?

Line 255: What is meant by showing a massive growth?

Figure 5C-H: What is the size of the scale bar? Are these images all at the same magnification?

Line 270: Is Figure 5E showing the stability of the nanoparticles after 3 months? This is not clear from the caption and the arrows drawn on the figure.

Line 275: Are the experiments with macrophages described in the methods or results?

Lines 281-282: Are these differences significant?

Line 292: What crosslinking was performed?